

# Incremental Analysis Update (IAU) in the Model for Prediction Across Scales coupled with the Joint Effort for Data assimilation Integration (MPAS-JEDI 2.0.0)

Soyoung Ha[1], Jonathan J. Guerrette[1,2], Ivette Hernández Baños[1], William C. Skamarock[1], and Michael G. Duda[1]

[1]National Center for Atmospheric Research, 3450 Mitchell Lane, Boulder, Colorado, USA
[2]Now at Tomorrow.io, Golden, Colorado, USA

**Correspondence:** Soyoung Ha (syha@ucar.edu)

**Abstract.** In a cycling system where data assimilation (DA) and model simulation are executed consecutively, the model forecasts initialized from the analysis (or data assimilation) can be systematically affected by dynamic imbalances generated during the analysis process. The high-frequency noise arising from the imbalances in the initial conditions can impose constraints on computational stability and efficiency during subsequent model simulations and can potentially become the low-frequency waves of physical significance. To mitigate these initial imbalances, the incremental analysis update (IAU) has long been utilized in the cycling context. This study introduces our recent implementation of the IAU in the Model for Prediction Across Scales for the Atmospheric component (MPAS-A), coupled with the Joint Effort for Data assimilation Integration (JEDI), through the cycling system called MPAS-Workflow. During the integration of the compressible nonhydrostatic equations in MPAS-A, analysis increments are distributed over a predefined time window (e.g., 6 hours) as fractional forcing at each time step. In a real case study with the assimilation of all conventional and satellite radiance observations every 6 h for one month, starting from mid-April 2018, model forecasts with IAU show that the initial noise illustrated by surface pressure tendency becomes well constrained throughout the forecast lead times, enhancing the system reliability. The month-long cycling with the assimilation of real observations demonstrates the successful implementation of the IAU capability in the MPAS-JEDI cycling system. Along with the comparison between the forecasts with and without IAU, several aspects on the implementation in MPAS-JEDI are discussed. Corresponding updates have been incorporated into the MPAS-A model (originally based on version 7.1), which is now publicly available in MPAS-JEDI and MPAS-Workflow Version 2.0.0.

## 1 Introduction

Data assimilation (DA) is a mathematical or statistical procedure to incorporate observations, unevenly distributed in time and space, into adjacent grids in the model forecast (or background) using relative weights based on the error statistics of the forecasts and observations. It does not consider dynamical or physical balances across model grids or variables, nor does it account for the conservation of mass, momentum, or energy. Hence, the initial balance of the atmospheric flow can by disrupted





by data assimilation when the initial state is replaced by the analysis state. Such imbalances can lead to artificial high-frequency noise, amplifying and propagating throughout the model simulation.

In the cycling system that alternates analysis and forecast, noise can continuously accumulate through cycles, which can degrade numerical stability and efficiency (with the time step smaller than $6\Delta x$ for a nominal grid spacing $\Delta x$). If the noise is not properly controlled and its nonlinear interactions with lower-frequency modes of physical interest are triggered, subsequent forecasts can be contaminated. If the forecast error growth is accelerated at each cycle, it not only limits the predictability of the atmospheric flow (Hohenegger and Schär, 2007), but can eventually cause the model simulation to crash.

To mitigate abrupt changes (or shocks) originating from inconsistencies or imbalances in the initial state, the incremental analysis update (IAU) method was introduced in Bloom et al. (1996) and has been widely used in leading operational or research centers (Polavarapu et al. (2004), Buehner et al. (2015), Lorenc et al. (2015), Lei and Whitaker (2016), Ha et al. (2017)). In this method, instead of initializing a numerical prediction model with the analysis state, analysis increments (e.g., analysis-minus-background) are distributed over a certain time window as fractional forcing applied to each time step during the model integration. Compared to other conventional continuous data assimilation approaches such as nudging, the IAU has an attractive time filtering feature, which can suppress spurious noise caused by data assimilation, providing a more consistent and balanced initial state before the next assimilation cycle.

The Model for Prediction Across Scales-Atmosphere (MPAS-A or MPAS, hereafter) utilizes centroidal Voronoi tessellations for its horizontal meshes and terrain-following height as a vertical coordinate, and employs a fully compressible nonhydrostatic model with the capability of high-resolution forecasting over a local area for both global and regional applications (Skamarock et al. (2012) and Skamarock et al. (2018)). Ha et al. (2017) first implemented the IAU in the MPAS Version 4 for a global cycling system with the Ensemble Kalman Filter (EnKF) in the Data Assimilation Research Testbed (DART; Anderson et al. (2009)) and demonstrated that it effectively suppressed spurious high-frequency noise, leading to better forecast skills. Since then, the model has been significantly upgraded for various new features, including acoustic filtering (Klemp et al. (2018)) that effectively damps out noise in the acoustic waves using horizontally explicit vertically implicit (HEVI) and split-explicit time integration schemes. The unstructured MPAS mesh can be configured with horizontally variable-resolution meshes that vary smoothly between low to high resolution regions, but the model integration time step is limited by the finest grid spacing of the mesh and uniformly applied across the domain to maintain the stability in the the fine mesh region.

Through the collaborative work with the Joint Effort for Data assimilation Integration (JEDI) team, an interface between MPAS and JEDI has been recently developed for a new community data assimilation system based on variational approach (Liu et al. (2022)). The MPAS-JEDI (or JEDI-MPAS) shares with other geophysical models the object-oriented framework for generic components for data assimilation such as the Interface for Observation Data Access (IODA; https://github.com/JCSDA/ioda, last access: 2 August 2023), the Unified Forward Operator (UFO; Honeyager et al. (2020), https://github.com/JCSDA/ufo, last access: 2 August 2023), the background error covariance models through System-Agnostic Background Error Representation (SABER; https://github.com/JCSDA/saber, last access:2 August 2023), and several minimization algorithms.

Since all of the analysis variables in MPAS-JEDI are either derived or diagnostic variables in the MPAS model, their analysis increments need to be converted back to the model's prognostic variables after the minimization so that the analysis updates can





be reflected in the model forecast. Variable transformations of mass fields are carried out based on the reconstructed pressure coordinate integrated from surface pressure, assuming hydrostatic balance and an ideal gas state. The recent version of MPAS-JEDI is updated to transform analysis increments to the increments of the model's prognostic variables instead of the full fields, as stated in Guerrette et al. (2023). This approach can reduce the errors arising from various assumptions or approximations made during variable transformations, but cannot avoid introducing imbalances between the prognostic fields and across the meshes due to the local nature of the observed quantity and the spatial localization applied to ensemble background error covariances, which can lead to potentially destructive effects on the quality of solutions from the compressible nonhydrostatic model solutions.

This article reports on our new implementation of the IAU feature in MPAS-JEDI using the cycling system called MPAS-Workflow (https://github.com/NCAR/MPAS-Workflow; last access:2 August 2023). For mathematical completeness, the IAU module in the MPAS model code (first released in V5.0) has been updated with several corrections. The main goal of the IAU is to stabilize the cycling system and maintain numerical efficiency by reducing imbalances in the initial state. But it will increase the forecast lead times because the IAU forcing is incorporated into the model prognostic equations every time step within a time window centered on the analysis time, followed by free forecasts. For 6-h cycling with a 6-h IAU time window, the total model integration time becomes 9 h, meaning the computation is increased by at least 30%. As a technical paper, default namelist parameters for IAU are listed, as defined in MPAS/namelist.atmosphere. This study does not discuss comprehensive characteristics of the complicated MPAS-JEDI or MPAS-Workflow, but focuses on the implementation of IAU, ensuring its reliability and performance through the forecast verification in surface pressure. Details of the implementation is described in Section 2, followed by the cycling system using MPAS-Workflow in Section 3. A real case study is presented in Section 4, then concluded in section 5.

## 2 The IAU implementation in MPAS-JEDI

The MPAS-A model uses a height-based coordinate following Klemp (2011), where terrain influences are progressively smoothed out toward the model top. The geometric height of coordinate surfaces ($z$) is defined by the combination of the nominal height (ignoring terrain) of coordinate surfaces ($\zeta$) and terrain height ($h_s$) on a two-dimensional (x,z) model domain

$$z = \zeta + A(\zeta)h_s(\mathrm{x},\zeta), \tag{1}$$

where $A(\zeta)$ = 1 - $\zeta/z_t$, indicating a weight for the terrain influence on the coordinate surfaces, and $z_t$ is the height of the model top. With $0 \leq A \leq 1 - \zeta/z_t$, an increasing amount of smoothing is applied at higher levels of $\zeta$ (e.g., decreasing $A(\zeta)$) such that the vertical coordinate transitions from terrain-following at the surface (e.g., $z = h_s(\mathrm{x},\zeta)$ for $\zeta = 0$) toward constant-height surfaces aloft. While rectangular grids are usually represented as (x,y), the horizontal unstructured mesh is defined solely by individual grid cells (in a random order). For simplicity, we denote the horizontal grid cell dimension as x, in addition to the model level ($z$), to specify the two-dimensional model dimension (x,z).

In the MPAS model, the nonhydrostatic compressible equation is formulated for conserved quantities (mass, momentum and moisture) represented in flux form using the split-explicit time integration techniques introduced in Klemp et al. (2007). As



described in Skamarock et al. (2012), defining a dry air density adjusted by a terrain transformation ($\tilde{\rho}_d = \rho_d/\zeta_z$, where $\rho_d$ is dry air density and $\nabla\zeta = (\zeta_x, \zeta_z)$), the flux variables X become

$$X = (\Theta_m, V_e, W, Q_j) = \tilde{\rho}_d \cdot (\theta_m, u_e, w, q_j), \tag{2}$$

where $\theta_m = \theta(1 + (R_v/R_d)q_v)$ as a potential temperature ($\theta$) modified by water vapor mixing ratio ($q_v$). $R_v$ and $R_d$ are gas constants for water vapor and dry air, respectively. $q_j$ represents the mixing ratio of each hydrometeor - water vapor ($q_v$), cloud
liquid water ($q_c$), cloud ice ($q_i$), rain ($q_r$), snow ($q_s$), graupel ($q_g$), and hail ($q_h$). The horizontal momentum is predicted in terms of the wind speed normal to cell edges ($u_e$), while $w$ stands for the vertical velocity. Coupled with $\tilde{\rho}_d$, all the flux-form variables in the prognostic equations now include terrain effects through transformation of the vertical coordinate. Note that all the variables associated with the vertical coordinate surfaces ($z$, $\zeta$ and $\nabla\zeta$) are constant fields.

The nonhydrostatic equations are integrated by updating total tendencies computed from each component of the model-
ing process. In the default configuration without IAU, the model is simply integrated from the initial condition updated with analysis variables at the initial time. However, if IAU is activated in the forecast (e.g., config_IAU_option = 'on' in namelist.atmosphere), instead of changing the initial state, we compute the analysis increments by subtracting the background forecast from the analysis and divide them by the total number of time steps within the IAU window for a three-dimensional IAU (3DIAU). In this context, the background forecast (or first-guess) is the forecast valid at the initial (or analysis) time from
the previous cycle. While the analysis file is produced by MPAS-JEDI (and is employed as a new initial condition for the model run without IAU), a separate analysis increment file at the analysis time ("AmB.nc") is created for the IAU forcing in MPAS-Workflow and provided to the model through another data stream called "iau". During the time integration, the total tendencies are adjusted by the IAU forcing ($\frac{\partial X}{\partial t}\big|_{amb}$) every time step as below.

$$\frac{\partial X}{\partial t}\bigg|_{total} = \frac{\partial X}{\partial t}\bigg|_{dyn} + \frac{\partial X}{\partial t}\bigg|_{phys} + \frac{\partial X}{\partial t}\bigg|_{amb}, \tag{3}$$

where the first and the second term in the right-hand side represent the total tendencies from dynamics and physics schemes, respectively.

In MPAS-JEDI, analysis variables are defined as temperature (T), specific humidity (s), surface pressure ($P_s$), zonal and meridional wind components (u and v, respectively) at cell centers by default. Except for surface pressure, all of them are two-dimensional (2D) variables in cells and levels (x,z). Note that in the unstructured mesh, 1-D indicates a horizontal plane,
while 2-D includes the vertical dimension (similar to a traditional 3-D Cartesian Coordinate). As none of these variables are prognostic in the model, once their increments ($\delta v = v^a - v^b$, where $a$ and $b$ stand for the analysis and background for the variable $v$, respectively) are computed through minimization, they are then transformed to the model's prognostic variables. As recently reported in Guerrette et al. (2023), analysis variables are updated in the incremental approach ($\phi^a = \phi^b + \delta\phi$), which requires the corresponding prior states ($\phi^b$) as well as the analysis increments ($\delta\phi$). The edge wind speed ($u_e$) is updated by
the increments in the horizontal wind components at cell centers (e.g., du and dv). Mass variable transformations begin with the increments in the approximated water vapor mixing ratio ($\delta q_v$), converted from the increments in specific humidity ($\delta s$) and its prior state ($s$). To linearize the equation $q_v = s/(1-s)$ for small increments $\delta q_v$ and $\delta s$, the first derivative of $q_v$ with



respect to $s$ is written as

$$q_v'(s) = \frac{d}{ds}(\frac{s}{1-s}) = \frac{1}{(1-s)^2}. \tag{4}$$

Using the first-order Taylor series expansion, we can linearize $q_v$ for the increments $\delta q_v$ and $\delta s$ as

$$q_v + \delta q_v \approx q_v(s) + q_v'(s)\delta s \approx \frac{s}{1-s} + \frac{1}{(1-s)^2}\delta s. \tag{5}$$

Then the increments in the pressure field are derived based on the changes in virtual temperature ($T_v = T(1 + 0.608\,q_v)$). Assuming hydrostatic balance, the two-dimensional pressure field ($P(x,z)$) is integrated upward from the surface based on the analyzed surface pressure ($P_s^a$). In the same incremental approach, dry air density ($\rho_d$) is updated based on the approximated

equation of state ($\rho_d = P/[R_d T_v (1 + q_v)]$), and potential temperature is derived from the relationship $\theta = T(\frac{P_0}{P})^{R_d/C_p}$, where $P_0 = 1000$ hPa is a reference pressure and $C_p$ the specific heat at constant pressure. Details are summarized in Appendix B.

While the nonhydrostatic MPAS-A model solves a prognostic equation for $\Theta_m = \tilde{\rho}_d \cdot \theta_m$ and diagnoses the full pressure ($p = p_0(R_d\zeta_z\Theta_m/p_0)^\gamma$, where $\gamma = C_p/C_v$), the analysis updates in MPAS-JEDI primarily rely on surface pressure (in both analysis and background) and several approximations, such as hydrostatic balance and the simplified moist conversion (e.g.,

ignoring all hydrometeors except for $q_v$).

As the dry density multiplied by the vertical coordinate Jacobian ($\tilde{\rho}_d$) is also updated during the analysis, once the prognostic variables are updated through the variable transformations, they should be re-coupled with the updated $\tilde{\rho}_d$, as in the right-hand side of Eq. (2), to compute the tendencies for the IAU forcing ($\frac{\partial X}{\partial t}\big|_{amb}$). When the model is configured without IAU, the recoupling step is carried out as part of the MPAS initialization process, but in the case of IAU, this is taken care of within the

IAU module in the MPAS model. The tendency for $\theta_m$ in the flux form, for instance, can be expressed in the partial equation

$$\frac{\partial X}{\partial t} = \frac{\partial}{\partial t}(\tilde{\rho}_d \cdot \theta_m) = \frac{\partial}{\partial t}(\tilde{\rho}_d \cdot \theta(1 + \frac{R_v}{R_d}q_v)). \tag{6}$$

Based on the differential rule, Eq.(6) can be rewritten as

$$\frac{\partial}{\partial t}(\tilde{\rho}_d \cdot \theta(1 + \frac{R_v}{R_d}q_v)) = \theta(1 + \frac{R_v}{R_d}q_v)\frac{\partial\tilde{\rho}_d}{\partial t} + \tilde{\rho}_d(1 + \frac{R_v}{R_d}q_v)\frac{\partial\theta}{\partial t} + \tilde{\rho}_d\theta\frac{\partial}{\partial t}(1 + \frac{R_v}{R_d}q_v) \tag{7}$$

$$= (1 + \frac{R_v}{R_d}q_v)(\theta\frac{\partial\tilde{\rho}_d}{\partial t} + \tilde{\rho}_d\frac{\partial\theta}{\partial t}) + \tilde{\rho}_d\theta\frac{R_v}{R_d}\frac{\partial q_v}{\partial t} \tag{8}$$

$$= (1 + \frac{R_v}{R_d}q_v)\frac{\partial(\tilde{\rho}_d\theta)}{\partial t} + \tilde{\rho}_d\theta\frac{R_v}{R_d}\frac{\partial q_v}{\partial t}. \tag{9}$$

The rightmost term in Eq.(9) contains $\tilde{\rho}_d\frac{\partial q_v}{\partial t}$, which can be derived from the tendency equation for water vapor mixing ratio, as below.

$$\frac{\partial}{\partial t}(\tilde{\rho}_d q_v) = \tilde{\rho}_d\frac{\partial q_v}{\partial t} + q_v\frac{\partial\tilde{\rho}_d}{\partial t} \tag{10}$$

$$\tilde{\rho}_d\frac{\partial q_v}{\partial t} = \frac{\partial(\tilde{\rho}_d q_v)}{\partial t} - q_v\frac{\partial\tilde{\rho}_d}{\partial t} \tag{11}$$



By incorporating Eq.(11) into Eq.(9), Eq.(6) can be rewritten as

$$\frac{\partial}{\partial t}(\tilde{\rho}_d \cdot \theta_m) = \frac{\partial}{\partial t}(\tilde{\rho}_d \cdot \theta(1 + \frac{R_v}{R_d}q_v)) = (1 + \frac{R_v}{R_d}q_v)\frac{\partial(\tilde{\rho}_d\theta)}{\partial t} + \theta\frac{R_v}{R_d}(\frac{\partial(\tilde{\rho}_d q_v)}{\partial t} - q_v\frac{\partial\tilde{\rho}_d}{\partial t}). \tag{12}$$

In the original IAU implementation in MPAS-A (Version 7.1 and prior), the last term $(-\theta\frac{R_v}{R_d}q_v\frac{\partial\tilde{\rho}_d}{\partial t})$ was omitted, but is now added back in the latest release of MPAS-JEDI. The missing term was a mathematical error, which needed to be fixed for accuracy. Our experiment on the coarse mesh with a nominal resolution of 120-km (as described in the following section) is not suitable for quantifying the error magnitude caused by this term. But there might be cases where the absence of this term can introduce sizable forecast errors in both thermodynamic and dynamic tendencies, especially in high-resolution simulations of strong pressure gradients.

The tendency for edge wind ($u_e$) can be described in the same manner, with the same correction applied.

$$\frac{\partial}{\partial t}(\tilde{\rho}_d \cdot u_e) = \tilde{\rho}_d\frac{\partial u_e}{\partial t} + u_e\frac{\partial\tilde{\rho}_d}{\partial t} \tag{13}$$

Here $u_e$ is coupled with dry air density ($\tilde{\rho}_d$) at edges, rather than cell centers. In the horizontally unstructured MPAS mesh, the center of each grid cell (mostly in a hexagonal shape) serves as the center of mass (centroidal), and cell edges bisect the lines connecting the two cell centers sharing the edges. Thus, the corresponding dry air density at the edge is defined as the mean of $\tilde{\rho}_d$ at the two cell centers that share the edge. (For the detailed description on the mesh characteristics, users refer to Fig.1 in Skamarock et al. (2012).) Note that all the model's prognostic variables except the edge wind ($u_e$) are defined at cell centers, coupled with $\tilde{\rho}_d$ at cell centers. While w is defined at the cell center of the horizontal mesh, it is C-grid staggered in the vertical.

In the IAU module, analysis increments ($\Delta x$) in the so-called "uncoupled" variables are read from the "iau" stream

$$\Delta x = x^a - x^b \quad where \quad x = (\rho_d, \theta, q_v) \tag{14}$$

so that the IAU tendencies ($\frac{\partial X}{\partial t}\big|_{amb}$) are computed for the coupled variables ($X$) as in Eqs.12 and 13. They are then multiplied by a predefined weighting function ($\omega_k$) to be applied every time step ($\Delta t$; config_dt) over the IAU time window ($\Delta\tau$; config_IAU_window_length_s).

$$\int_{-\frac{\Delta\tau}{2}}^{\frac{\Delta\tau}{2}} \frac{\partial X}{\partial t}\bigg|_{amb} dt = \sum_{k=1}^{n}\omega_k \cdot \frac{\Delta X}{\Delta t} = \sum_{k=1}^{n}\omega_k \cdot \frac{\Delta(\tilde{\rho}_d \cdot x)}{\Delta t}, \tag{15}$$

where $\Delta X = X^a - X^b$, $n = \Delta\tau/\Delta t$ and $\omega_k = 1/n$ for each time step k. In the default configuration for 3DIAU, the IAU forcing computed from analysis increments is evenly distributed across time step for the 6-h IAU window, followed by 3-h free forecast to the next analysis time, making a total of 9-h background forecast at each analysis cycle (Fig. 1). Although a simple 3DIAU is currently implemented with constant forcing, it is easily extended for 4DIAU with varying weights over the IAU time window.



## 3  Cycling data assimilation with MPAS-Workflow

Our open-source MPAS-Workflow was introduced in Guerrette et al. (2023) and has only been tested on Cheyenne, one of the National Center for Atmospheric Research (NCAR)'s High-Performance Computers (HPCs), but we describe some technical details here since it is at the heart of the cycling system for MPAS-JEDI and it has gone through major updates for the IAU feature. The MPAS-Workflow uses the Cylc general purpose workflow engine (v7.8.3) (Oliver et al. (2019), https://cylc.github.io/; last access: Aug 29, 2023), and it is designed for end-to-end processes of the MPAS-JEDI cycling system. It controls all the parameters for running the MPAS model and data assimilation, with high flexibility for a number of different configurations. It defines various lists of variables, such as those for the analysis, background, and observation types to assimilate. Furthermore, it is equipped with a Python-based post-processing package (https://github.com/JCSDA/mpas-jedi/tree/2.0.0/graphics; last access: Aug 2, 2023), including diagnostics and plotting utilities.

Once IAU is activated in MPAS-Workflow, the model initial and run times are automatically adjusted to -3 h and 9 h, respectively, for a 6-h IAU time window, as depicted in Fig.1. Also, the analysis increment file ("AmB.nc") is created at the analysis time (e.g., t=0), while the background forecast valid at -3 h (instead of 0 h) is employed from the previous cycle. Due to the availability of the first-guess file, the IAU option is activated only from the second cycle, and the model output interval is changed from 6 h to 3 h for 6-h cycling.

MPAS-JEDI employs its own customized version of the MPAS-A model, using a two-stream I/O approach by default, to run DA cycling more efficiently. The two-stream (or split) I/O approach was originally developed for DA cycling in a restart mode to avoid writing time-invariant fields in every restart file while ensuring the model forecasts reproducible. In the restart mode, the MPAS-A model produces a restart file with about 230 variables, among which only ~70 variables vary with time, while the rest of them are time-invariant. Note that these static fields are not the same as those in the static file for the MPAS initialization because the MPAS static file only contains horizontal fields, with no vertical dimension or coordinate. This is because the MPAS initialization consists of two consecutive steps - the first step for constructing the 1-D horizontal mesh in the static file, and the second for producing 2-D fields based on the terrain-following height vertical coordinate in the initial condition file. By splitting the restart file for the variables between time-variant and time-invariant, we can cycle with a much smaller file containing time-variant fields only, which is approximately 1/6 of the original size of the restart file. We refer to the new I/O stream as "da_state" in the MPAS-A model, as the file serves as both input and output for DA cycling and can be updated by the analysis process. In fact, this I/O stream can be used in a restart mode regardless of DA or DA cycling. One caveat of the split I/O approach is that the reproducibility might depend on the model configuration (and potentially the version), meaning that the variable list of "da_state" is not always applicable to or guaranteed for all different namelist options for MPAS-A. We had initially developed this new I/O stream in the MPAS-A model based on version 6.1 and ensured its bit-for-bit reproducibility. However, the MPAS-JEDI cycling system is run in a cold-start mode, initializing all the physics tendencies at the analysis time, so it only uses the "da_state" stream as a shorter version of the model input and output files, not to replace a restart file. At the time of writing, this new I/O stream is available only in MPAS-JEDI, but it will be merged into the official version of the MPAS-A model (https://github.com/MPAS-Dev/MPAS-Model) in the future.



The variational approach essentially linearizes the model and constructs a static background (or forecast) error covariance to find an analysis solution closest to observations iteratively (e.g., through a minimization process). Although the static background error covariance only represents the climatological information (e.g., with no temporal variations), it is a key component

of variational data assimilation algorithms, modeling the relationships between control variables through physical transformation or balance operators as well as spatial auto-correlations of each control variable to determine how to propagate the observed information across model grids and variables (Descombes et al. (2015)). In this study, we use a pure ensemble-variational (EnVar) approach with the zero static error covariance. The ensemble background error covariance composed of 20-member 6-h MPAS forecasts initialized from the NCEP's 20-member Global Ensemble Forecast System (GEFS) ensemble analysis at each

cycle, like in Liu et al. (2022).

In the JEDI system, the Unified Forward Operator (UFO) not only provides observation operators (named "HofX") to compute innovations (e.g., differences between observations and the corresponding forecasts; (o-f)'s) for all different observation types, but also handles all the data quality control (QC) filters, data range or coverage area, data manipulation (such as data thinning or averaging onto model levels), bias correction, and the specification of observation errors (including observation

uncertainties and error correlations, if applicable). Due to various filters (or QCs) that can be applied to observations, the UFO produces observation errors before and after the QCs, named "ObsError" and "EffectiveErrors", respectively. While "ObsError" indicates the initial observation error values from the input observation file, the actual observation errors applied to data assimilation can be found in "EffectiveErrors". When multiple filters are applied to observations or observation operators (such as data thinning or variable transformation), users can specify the order of filters through the YAML configuration files. The

default quality control (called "PreQC") rejects observations when data QC flag is larger than 3, as provided by input observation files (such as GSI-ncdiag files used in this study). In the assimilation of surface observations, for example, they are also discarded if innovations exceed three times the standard deviation of the observation error ($\sigma_o$), or if the station height differs from the model's surface elevation by more than 200 m. In this study, the height difference threshold is reduced to 100 m and the surface pressure terrain height correction (called "SfcPCorrected") uses the so-called WRFDA method rather than

the default UKMO method (Ingleby (2013)). In our initial test (not shown), the difference between different height correction methods was insignificant, but it would be worth revisiting in the future, especially for regions with significant orography. While the common functions or modules are located under UFO in JEDI, most of these options are controlled through YAML configurations in MPAS-Workflow. An example for surface data assimilation is provided in appendix.

## 4 Experiments

After the new implementation of IAU in MPAS V7, global analysis and forecast cycling was conducted over a global 120-km quasi-uniform mesh every 6 h for one month, starting from April 15, 2018, using the MPAS-Workflow for the hybrid 3DEnVar in the MPAS-JEDI system. During the cycling, all the conventional observations, satellite winds, and clear-sky microwave radiances from 6 Advanced Microwave Sounding Unit-A (AMSU-A) sensors aboard NOAA-15, NOAA-18, NOAA-19, AQUA,





METOP-A, and METOP-B were assimilated together, using diagonal observation error covariances and a pure ensemble back-
ground error covariance (computed from GEFS), like in Guerrette et al. (2023).

In the model simulation, a "mesoscale_reference" physics suite is used that includes WSM6 microphysics, new Tiedtke
cumulus, YSU PBL, YSU gravity wave drag over orography (GWDO), RRTMG SW and LW, Noah LSM. Ozone climatology
is activated, and radiation effective radii for cloud water ($q_c$), cloud ice ($q_i$), and snow ($q_s$) are computed in the microphysics
scheme (e.g., config_microp_re = true).

Two month-long cycling experiments are conducted with and without IAU, named IAU and CTRL, respectively. Figure 2
illustrates a comparison of the absolute value of the surface pressure tendency ($|\frac{dP_s}{dt}|$) as an area-weighted global mean, in the
background forecasts from the analysis (shown at zero on the x-axis), valid at 00 UTC May 1, 2018. Even after a two-week
spinup period, it shows that the initial noise arising from the analysis increments is very high in the control run ("CTRL" in
dashed line), but the IAU effectively suppresses such noise throughout the 9-h forecast. The asymptotic value is ∼0.013 [Pa
s$^{-1}$] for 9-h forecast, which is comparable to ∼0.01 [Pa s$^{-1}$] for 6-h forecast, as presented in Lynch and Huang (1992) in their
digital filter initialization (DFI) study. In Fig. 3, the horizontal distribution reveals that noise from the initial state in CTRL is
widespread across the globe from the first time step, whereas it almost disappears with IAU. It is noted that, in the cycling with
IAU, the forecast starts from -3 h, leading to a different initial time for model integration compared to the control run (which
starts at the analysis time, i.e., 0 h). However, regardless of the actual initial time, our focus here is on comparing the deviation
of the first time step from the initial state.

In DA cycling, it is common to monitor the total number of observations assimilated at each cycle. A time series with a
declining trend might be indicative of the analysis in a poor quality, rejecting more observations with cycles. As shown in
Figure 4, the positive differences in (IAU - CTRL; red) indicate that slightly more observations are assimilated with IAU. The
difference is small compared to the total number of observations (in gray line with the right y-axis), but IAU tends to assimilate
more observations in most cycles, another sign of our successful implementation.

To examine the performance of cycling DA, Fig. 5 compares observation-minus-background (omb) in (a) root-mean-square
(rms) errors and (b) mean errors over the globe for each cycle. Note that omb is computed for 6 h background forecasts in
CTRL, but for 9 h forecasts in the IAU run, both at the same validation time. The rms errors in IAU (dots in red line) are
slightly yet consistently smaller than those in CTRL ('+' in black line) throughout the cycling period. In terms of mean errors,
however, the month-long average in surface pressure is consistent at 20 Pa for both experiments. The time series illustrates
that both rms and bias errors initially start with large error magnitudes, but after about one week of cycling, the errors tend to
stabilize. While the global-mean rms error in CTRL, averaged over all the cycles following the one-week spin-up period, is
∼1.2 hPa, IAU leads to an improvement in the forecast error by approximately 3% (e.g., (CTRL-IAU)/CTRL x 100 = -3%).

In the sounding verification over the globe, the percentage difference of rms errors in IAU with respect to the one in CTRL
for the entire cycling period also shows slight but systematic improvements in the background forecasts, as depicted in Fig.6.
The impact on zonal wind is almost neutral, but temperature forecasts near the surface (e.g., 1000 hPa) are improved by 3.6%,
similar to the improvement in surface pressure.



We also run 5-day forecasts from the 00 UTC analysis every day and compute forecast errors with respect to the GFS analysis. Fig.7 displays rms errors in CTRL for 5-day forecasts on the x-axis across latitudes (on the y-axis) on the right, while

the relative differences in rms errors in IAU (%) are depicted on the left, with red indicating degradation and blue implying forecast improvements over the baseline (CTRL). Statistical significance is denoted by light circles at the 95% confidence level. Compared to the GFS analysis, MPAS forecasts in CTRL exhibit the largest (or the fastest) error growth in the Southern Hemisphere. Forecasts in the IAU run, on the other hand, tend to reduce errors in the Tropics and Southern Hemisphere while increasing errors near the North Pole region. As the GFS analysis also suffers from its own errors (Bhargava et al. (2018)), the

forecast verification against the analysis is not intended for a thorough investigation on the model performance. It rather serves to introduce the post-processing capabilities in both model and observation spaces provided by MPAS-Workflow.

The overall results from the cycling experiments are promising, showing the reliability throughout the month-long period. We can examine the impact of the IAU option through more extensive diagnostics against other observation types and variables, and through the evaluation of longer forecasts in the future. As a proof of concept, only 3DIAU on a 120 km global mesh was

tested here, but it was implemented in a way to make it extensible with different weighting functions or even to 4DIAU in the MPAS model. Also, it is applicable to variable-resolution meshes as it is in case one would want to examine the impact of IAU over the area with mesh refinement.

## 5 Conclusions

This study introduces the incremental analysis update (IAU) implemented in the MPAS-JEDI cycling system operated by

MPAS-Workflow. Through a real case study for one month, starting from mid-April 2018, assimilating all conventional, aircraft, and satellite radiance observations, we demonstrate that the IAU is successfully implemented, effectively suppressing the artificial noise produced by initial imbalances during the analysis process. Although the current implementation is a simple three-dimensional IAU (3DIAU) with the same fractional forcing applied to each time step, there are several aspects that might be worth pointing out in regards to our development effort in MPAS-JEDI: i) computational stability and efficiency might be

critical to any numerical weather prediction (NWP) models, but special attention was taken to the MPAS-A model which solves the compressible nonhydrostatic equations employing an unstructured mesh based on centroidal Voronoi tessellations. In the model integration, a time step is set based on the smallest grid spacing of a given unstructured mesh, then uniformly applied across the entire mesh (e.g., regardless of the nominal grid spacing of individual grid cells). To ensure numerical stability even for the unstructured mesh applications, various filtering techniques are carefully designed and applied to the model numerics

(Klemp et al. (2007, 2018)). Aside from the modeling approach, the IAU is considered another efficient way of controlling high-frequency oscillations produced by the analysis procedure so that energy does not accumulate in the acoustic or unbalanced gravity wave modes through cycles due to the initialization. ii) the MPAS-A model treats prognostic variables in the flux-form, meaning that the variables are coupled with dry air density. Because the density is also updated as part of the analysis variables, analysis increments in the IAU forcing term should also be computed in the tendency form of each analysis variable recoupled

with the updated density. During our implementation, the IAU module is corrected to properly represent changes in the density



from the analysis. iii) analysis increments in MPAS-JEDI are basically computed from the linearized version of the hydrostatic balance equation that is vertically integrated from the analyzed surface pressure. Horizontally, MPAS-JEDI updates all the analysis variables in the model's native (e.g., unstructured) mesh, but vertically the MPAS-A height coordinate is mapped to the hydrostatic pressure coordinate for the increments based on the surface pressure analysis. A global 120-km mesh employed

for our cycling experiments demonstrates our technical implementation of the IAU capability in MPAS-JEDI. As we move towards convective-scale data assimilation with unconventional observations such as radars or lidars, however, it would be worth revisiting the influence of IAU on either variable- or higher-resolution mesh, in conjunction with various DA techniques available in MPAS-JEDI.

*Code and data availability.* All the codes and scripts are available on the public GitHub domain with free access. The MPAS-JEDI can be

downloaded from https://github.com/JCSDA/mpas-bundle (release/2.0.0 branch). MPAS-Workflow scripts are available in https://github.com/NCAR/MPAS-Workflow.



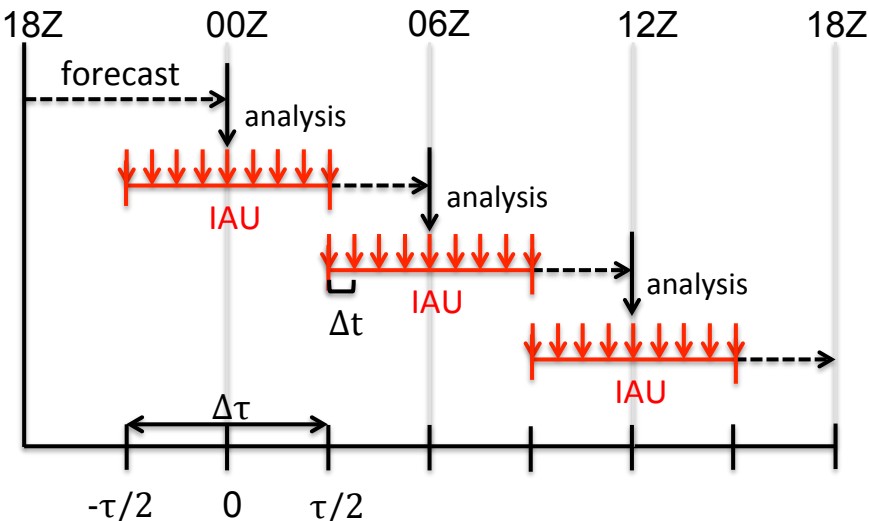

**Figure 1.** A diagram for cycling with IAU. $\Delta\tau$ is the IAU time window (config_IAU_window_length_s) and $\Delta t$ an integration time step (config_dt) defined in namelist.atmosphere.

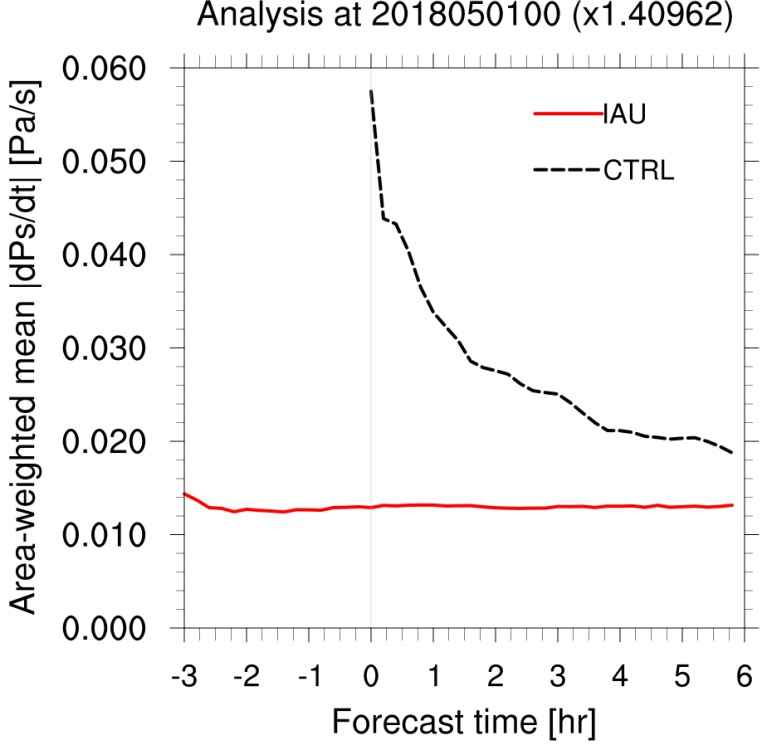

**Figure 2.** Time series of the globally averaged absolute value of the surface pressure tendency ($|dP_s/dt|$) in the forecast from the 00 UTC May 1, 2018 analysis over the 120-km quasi-uniform mesh.



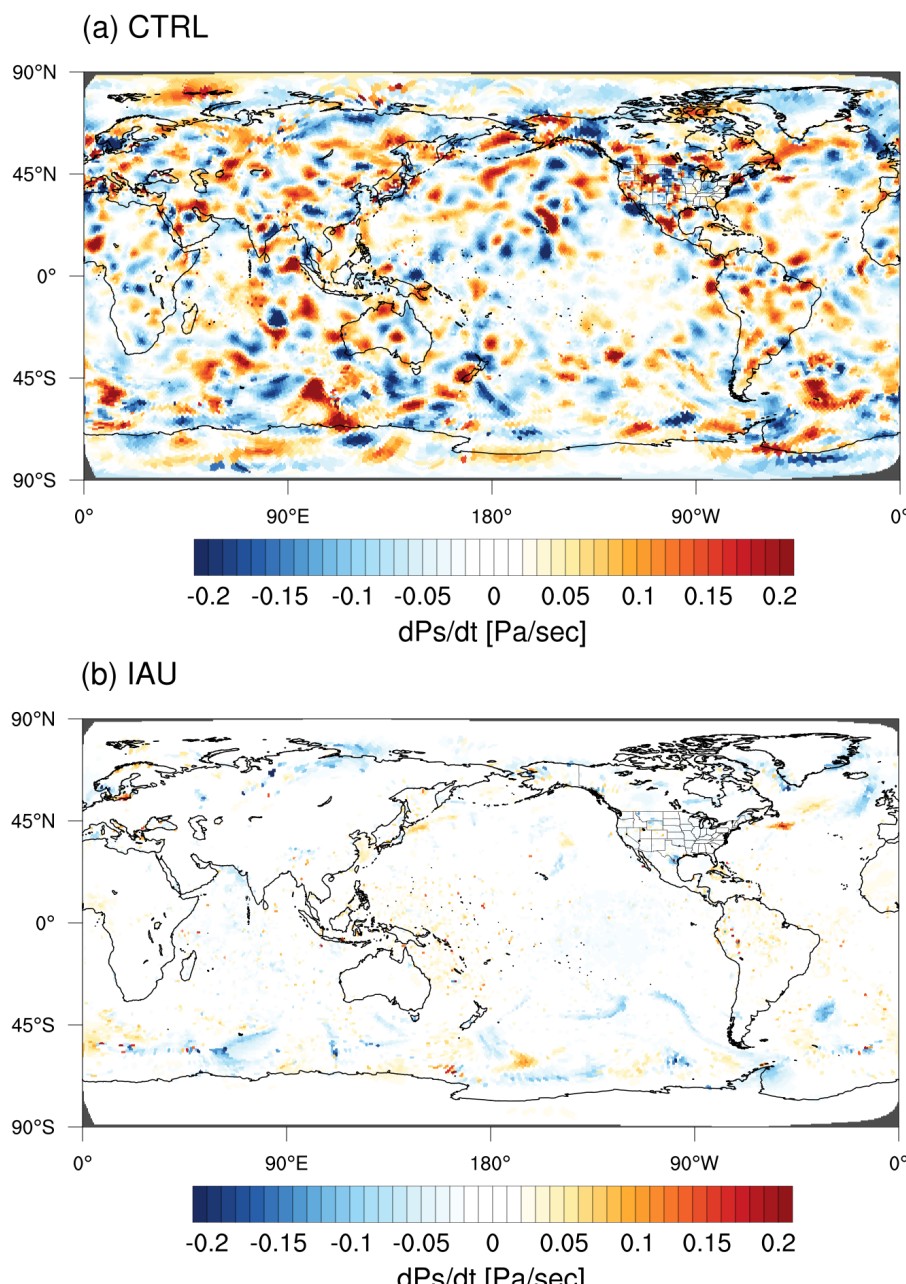

**Figure 3.** Horizontal distribution of $dP_s/dt$ simulated in the first time step from the initial conditions in (a) the control run without IAU and (b) the IAU run.



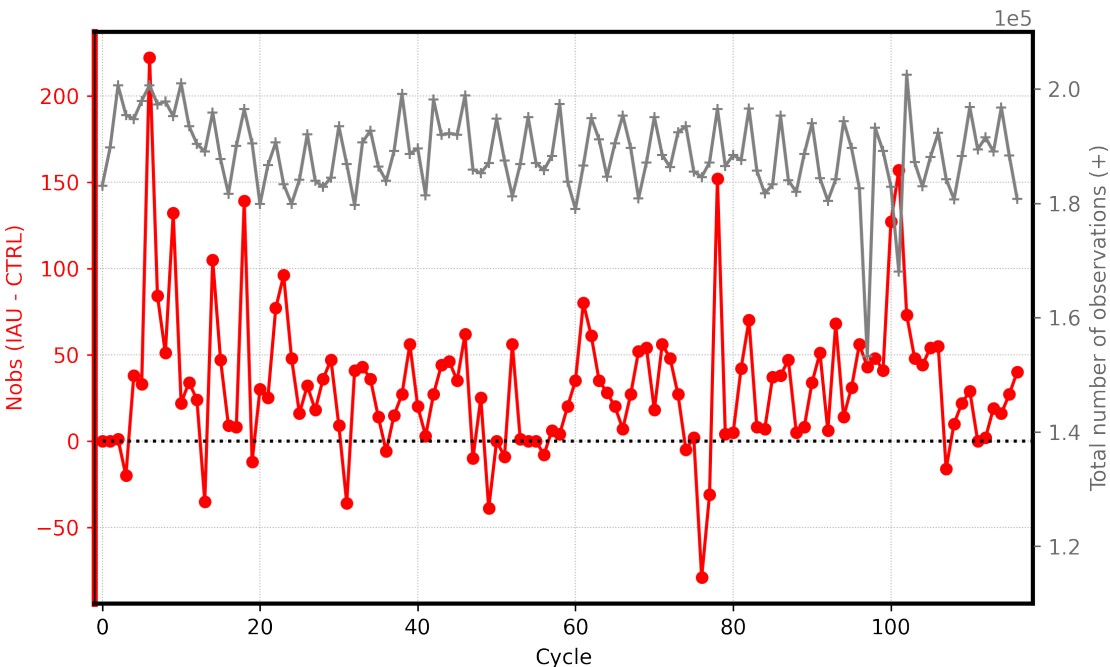

**Figure 4.** Time series of the difference in the total number of surface pressure observations assimilated in CTRL and IAU (red), and the total number of the observations available at each cycle (gray).

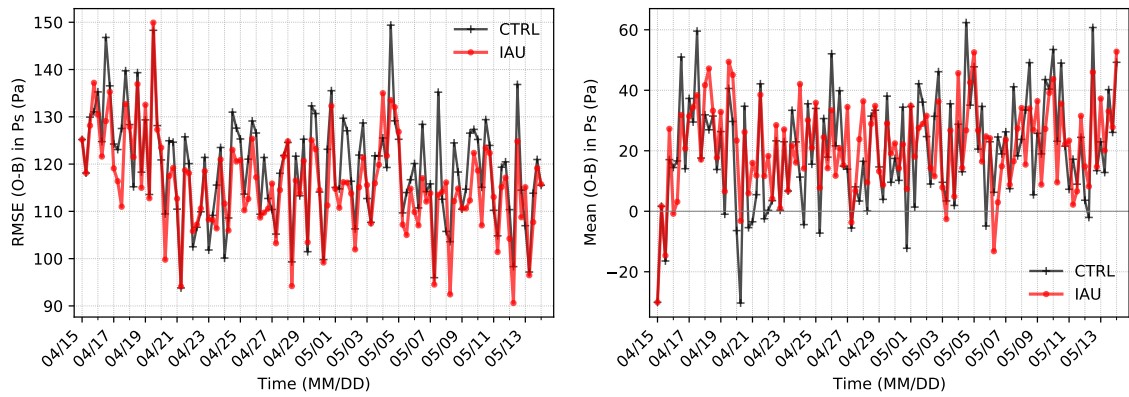

**Figure 5.** Time series of (left) root-mean-square (RMS) and (right) mean bias errors of surface pressure (Pa) in (o-b)'s in both CTRL and IAU runs during the cycles.





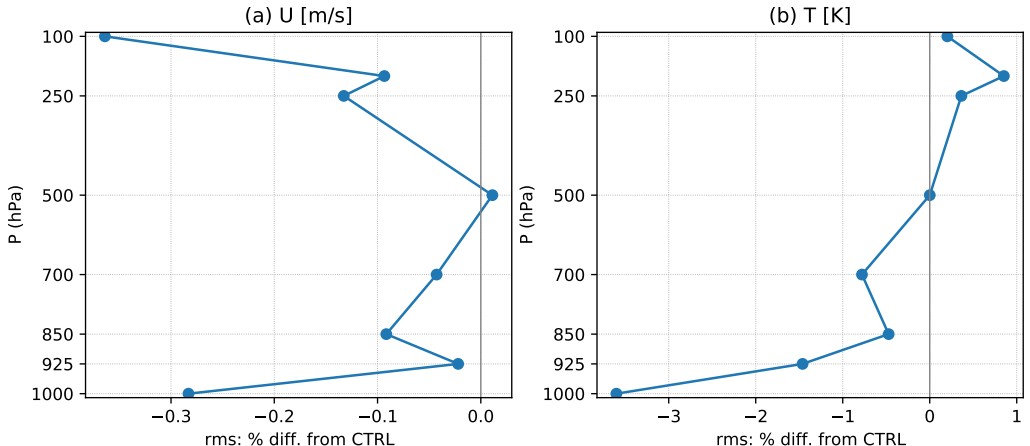

**Figure 6.** Percent difference of RMS (o-b)'s in IAU with respect to the one in CTRL for radiosonde verification in (a) zonal wind (m/s) and (b) temperature (K).

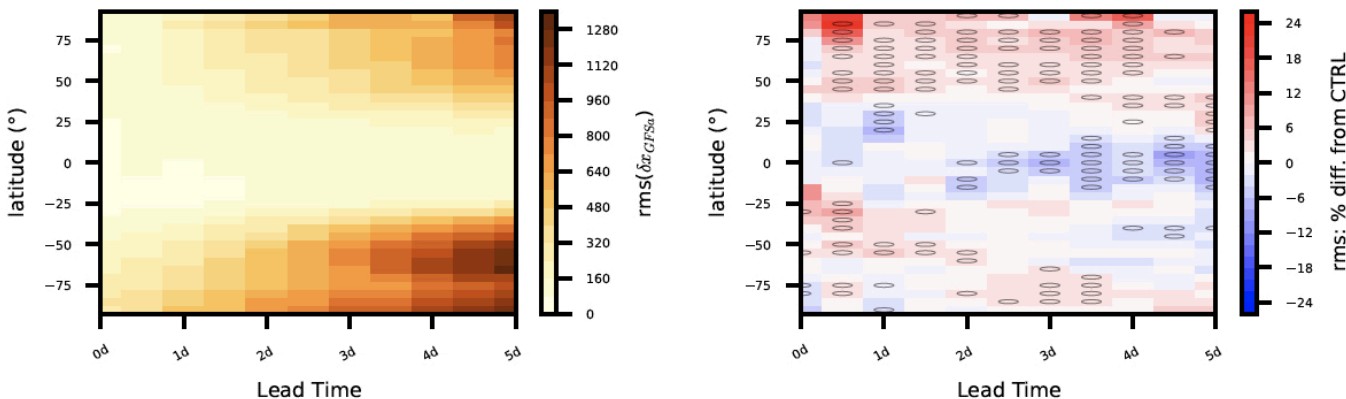

**Figure 7.** The rms errors of 5-day forecasts in surface pressure at different latitudes relative to the GFS analysis are displayed for CTRL (left) and as the rms differences from CTRL in percentages in IAU (right) with statistical significance at the 95% confidence level indicated by light circles.



**Appendix A: Cycling with MPAS-Workflow**

MPAS-JEDI is an interface between the MPAS-A model and the JEDI data assimilation system, including all the model-specific components such as variable transformation and HofX that computes analysis increments in the MPAS variables. To cycle MPAS-JEDI over a period of time in a synthetic way, MPAS-Workflow controls the entire data streams as well as all the configurations for data assimilation and model forecasts. As IAU changes the input/output file stream and the model configuration, they are all taken care of in MPAS-Workflow (mostly through various YAML configurations).

The MPAS-Workflow offers high flexibility for a number of applications such as data assimilation with 3DVar, pure 3/4DEnVar, and hybrid 3DEnVar with dual resolution, ensemble data assimilation, and more recently with the Local Ensemble Transform Kalman Filter (LETKF) algorithm in MPAS-JEDI. In addition, it can generate observations in the IODA format, the GFS analyses in MPAS initial conditions format, and free deterministic forecasts from the GFS analyses using specific Cylc suites (no cycling). Observations and GFS analyses can be obtained from the NCAR Research Data Archive (RDA) or the NCEP FTP server. With all of these capabilities, it has been tested for near real-time cycling runs using the 3DVar algorithm.

At the time of writing, the MPAS-Workflow does not build either MPAS or JEDI, which should be built separately after downloading the source codes from https://github.com/JCSDA-internal/mpas-bundle/ (release/2.0.0 branch). The mpas-bundle is built using cmake, a set of CMake macros provided by the European Centre for Medium-Range Weather Forecasts (ECMWF), along with their libraries. For the installation guide for those tools, readers can refer to https://jointcenterforsatellitedataassimilation-jedi-docs.readthedocs-hosted.com/en/latest/inside/developer_tools/cmake.html. Once built, the path should be specified in initialize/framework/Build.py in the MPAS-Workflow.

To run a cycling experiment with IAU, all users need to do is to edit a single YAML file. Each section controls the specific configuration to build up the YAML file of the MPAS-JEDI application (i.e., pure 3DEnVar) and other components of the workflow to construct the Cylc suite that will orchestrate the cycling experiment. For IAU, a new logical parameter has to be added in a new line as "IAU: True". Here is the configuration employed this study.

<MPAS-Workflow/scenarios/3denvar_OIE120km_WarmStart_IAU.yaml>

```
workflow:
first cycle point: 20180414T18
final cycle point: 20180510T00
experiment:
suffix: '_IAU'
observations:
resource: PANDACArchive
members:
n: 1
model:
outerMesh: 120km
```



```
    innerMesh: 120km
    ensembleMesh: 120km
    firstbackground:
    resource: "PANDAC.GFS"
externalanalyses:
    resource: "GFS.PANDAC"
    variational:
    DAType: 3denvar
    ensemble:
forecasts:
    resource: "PANDAC.GEFS"
    forecast:
    IAU: True
```

The YAML file used to run MPAS-JEDI with 3DEnVar is built by adding the observers snippets to the application sections (see MPAS-Workflow/config/jedi/applications/3denvar.yaml). Here we also provide two more sample YAML configurations used in this study for assimilating surface observations.

1) MPAS-Workflow/config/jedi/ObsPlugs/da/filters/sfc.yaml (for Data QC)

```
    obs filters:
- filter: PreQC
    maxvalue: 3 ♯ Maximum data QC flag
    - filter: Difference Check
    reference: MetaData/stationElevation
    value: GeoVaLs/surface_altitude
threshold: 100.0 ♯ default: 200.
    - filter: Background Check
    threshold: 3.0
```

2) MPAS-Workflow/config/jedi/ObsPlugs/da/base/sfc.yaml (Options for an observation operator for surface observations.)

```
- obs space:
    name: SfcPCorrected
    _obsdatain: &ObsDataIn
    engine:
    type: H5File
obsfile: InDBDir/sfc_obs_thisValidDate.h5
```



```
_obsdataout: &ObsDataOut
engine:
type: H5File
obsfile: OutDBDirMemberDir/obsPrefix_sfcObsOutSuffix.h5
obsdatain: *ObsDataIn
ObsDataOut
simulated variables: [stationPressure]
obs error: *ObsErrorDiagonal
obs operator:
name: SfcPCorrected
da_psfc_scheme: WRFDA ♯ default: UKMO
linear obs operator:
name: Identity
observation alias file: obsop_name_map.yaml
```

## Appendix B: Linearized equations for incremental variable transformations

As described in Section 2, MPAS-JEDI first computes the analysis through the iterative minimization procedure, then converts the increments in the analysis variables to the model's prognostic fields. Based on Eq.(5), the increments in water vapor mixing ratio are computed as $\delta q_v^k = \delta s^k/(1-s^k)^2$ at each model level k. Then, the increments in virtual temperature ($\delta T_v^k$) and pressure ($\delta P^k$) are derived using the first derivative as follows.

$$\delta T_v^k = \delta T^k(1 + 0.608\,q_v^k) + 0.608 \cdot \delta q_v^k \cdot T^k \tag{B1}$$

$$\delta P^k = \delta P^{k-1}\exp\{-\frac{g(z^k - z^{k-1})}{R_d T_v^k}\} + P^{k-1}\exp\{-\frac{g(z^k - z^{k-1})}{R_d T_v^k}\}\frac{g(z^k - z^{k-1})}{R_d T_v^{k^2}}\delta T_v^k, \tag{B2}$$

where $R_d$ is the gas constant for dry air. With $P^0 = P_s$ and $z^0 = z_s$ (e.g., terrain height) at the lowest level k = 1, the equations are applied to each model level upward from the surface.

The increments in dry air density ($\delta\rho_d$) and potential temperature ($\delta\theta$) are also derived at each level through the linearized formulas below. (As all the variables are computed at the same level k, we omit the superscript k.)

$$\rho_d + \delta\rho_d \approx \rho_d + \frac{\partial\rho_d}{\partial P} \cdot \delta P + \frac{\partial\rho_d}{\partial T_v} \cdot \delta T_v + \frac{\partial\rho_d}{\partial q_v} \cdot \delta q_v$$
$$\approx \frac{P}{R_d T_v(1+q_v)} + \frac{1}{R_d T_v(1+q_v)} \cdot \delta P - \frac{P}{R_d T_v{}^2(1+q_v)} \cdot \delta T_v - \frac{P}{R_d T_v(1+q_v)^2} \cdot \delta q_v$$



$$\theta + \delta\theta \approx \theta + \frac{\partial\theta}{\partial T} \cdot \delta T + \frac{\partial\theta}{\partial P} \cdot \delta P$$

$$\approx T\left(\frac{P_0}{P}\right)^{\frac{R_d}{C_p}} + \left(\frac{P_0}{P}\right)^{\frac{R_d}{C_p}} \cdot \delta T - \frac{R_d}{C_p}\frac{T}{P}\left(\frac{P_0}{P}\right)^{\frac{R_d}{C_p}} \cdot \delta P$$

Here, the first term on the right-hand side (RHS) is replaced with the background, and the full fields (e.g., variables without $\delta$ in front of them) represent the prior states as well.

*Author contributions.* SH implemented the IAU in the MPAS-JEDI, conducting experiments, analyzing the results, and writing the manuscript. JG updated the MPAS-Workflow for the IAU capability. IB helped producing plots and edited Appendix A comprehensively. WS and MD worked with SH on the implementation of IAU and the two-stream I/O in the MPAS-A model. All the co-authors edited the manuscript.

*Competing interests.* Authors declare that there are no competing interests.

*Acknowledgements.* This work was jointly funded by the United States Air Force (Grant no. NA21OAR4310383) and the National Science Foundation under Cooperative Agreement No. 1852977.





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
