# Peer review of "Incremental Analysis Update (IAU) in the Model for Prediction Across Scales coupled with the Joint Effort for Data assimilation Integration (MPAS-JEDI 2.0.0)"

_EGUsphere, 2023_

## Author Comment (AC2)

==============================
Response to anonymous referee #1:
Citation: https://doi.org/10.5194/egusphere-2023-2299-RC1

Thank you very much for your kind review. We believe that your comments have greatly improved the clarity of our draft. Regarding the last two scientific questions, we would like to remind you that this article was submitted to the manuscript type of "Development and technical papers" (https://www.geoscientific-model-development.net/about/manuscript_types.html). Hence, we focused on our new implementation, demonstrating its successful development. We'd appreciate your understanding and support in letting us stay focused. Please find our point-by-point response to your valuable comments below. All our responses are marked in blue.

This manuscript introduced the three-dimensional IAU implemented in the MPAS-JEDI 2.0.0. Previous studies have shown that IAU can effectively remedy the imbalance caused by intermittent data assimilation. It is worthwhile to investigate the performances and potential issues of IAU for a global model with varying horizontal meshes, which would provide guidance for future seamless predictions. The manuscript is pleasant to read. Please see my specific comments as below.

l58-60, this sentence needs be clarified. What's the difference between full fields and prognostic variables? Are the prognostic variables subsets of the full fields? If yes, why transform to the prognostic variables imposes more imbalances than that to full fields?

⇨ The reviewer seems to misinterpret the statement in L58-59 "The recent version of MPAS-JEDI is updated to transform analysis increments to the increments of the model's prognostic variables instead of the full fields". Here, we meant updating the increments ($dv$) rather than the full values ($v$) for each prognostic variable, not differentiating full vs. prognostic variables.

⇨ But to improve the clarity, we've now updated the statement as "The recent version of MPAS-JEDI is updated to transform analysis increments to the increments of the model's prognostic variables (instead of their full fields), as stated in Guerrette et al. (2023b)."

⇨ Also, in new L115-121, we added "As defined in Eq. (2), the MPAS model only predicts 2-D $\tilde{\rho_d}$, $\theta_m$, $u_e$, $w$, and hydrometeors in mixing ratios. In other words, none of the analysis variables are prognostic, meaning that once their increments ($\delta v = v^a - v^b$, where $a$ and $b$ stand for the analysis and background for the variable $v$, respectively) are computed through minimization, they should be transformed to the prognostic variables for model integration. To reduce potential errors resulting from approximations such as hydrostatic balance and the equation of state, variable transformations are only applied to the increments ($\delta{v}$) rather than the full analysis fields ($v^a$), keeping prior states ($v^b$) as nonhydrostatic forecasts from the previous cycle."

l62-64, it would be nice to add some references for the imbalances mentioned here.

⇨ Please note that references were already made for IAU as a way to identify and address imbalances in L29-32. Also, reference to Ha et al. (2017) was also made in L40-42 for

imbalances with EnKF in MPAS-DART. To the best of our knowledge, this manuscript is the first of its kind to discuss such imbalances in the MPAS-JEDI system, using the incremental approach within a pure 3DEnVar algorithm. Therefore, we hope this study can serve as another reference for future studies on the imbalances associated with DA. No changes are made.

l118, what is phi?
- ⇨ Sorry for the confusion and thanks for catching this up! It is just representative of a model variable.
- ⇨ The whole paragraph is now updated (in new L115-121) and uses $v$ to make it consistent with the expression in L117 (e.g., $\delta v = v^a - v^b$).

l112-122, since the transformation from grid point to mesh grid is not linear, it is not equivalent to transform the increment or the analysis. For the MPAS-IAU, is the native increment or the analysis used as the input for MPAS simulations?
- ⇨ It is unclear what the reviewer meant by the transformation from grid point to mesh grid, since there was no such a thing here. Presumably, there might be a misunderstanding regarding our variable transformations. The reviewer previously asked about L58-60, expressing similar confusion. It is noted that we are always working with the model's native (e.g., unstructured) mesh and the variable transformations are unrelated with the mesh. Just to be clear, we are only converting one variable to another located on the same mesh location and variables are always treated in the native mesh. Since the mesh info for the variables is already described in L114-116, no more changes.

l127-130, how much the error could be introduced by this hydrostatic assumption?
- ⇨ Errors would depend on various factors such as horizontal and vertical mesh resolutions and flow regimes in the simulation. Convective storms, for instance, could be highly nonlinear and nonhydrostatic, but the errors arising from variable transformations alone are not yet quantified. As MPAS-JEDI is being actively developed for convective-scale DA, the potential issue could be examined in future studies. No changes.

l176, please spell 4DIAU out at the first time.
- ⇨ Thank you. The statement is now updated as "Although a simple 3DIAU is currently implemented with constant forcing, it could be extended for a four-dimensional IAU (4DIAU) with varying weights over the IAU time window."

l173-177, it is interesting to know the IAU terms for hydrometer variables. Are they the same as Eq. 12?
- ⇨ As stated in L94-95, water vapor mixing ratio (q_v) is part of hydrometeors. The tendency expressed for q_v in Eq. (10) can be applied to any q_j. No changes.

l195-210, if a restart file is used for cycling, how the analysis and analysis tendency are computed for multiple time slices?

⇨ The frequency of DA is unrelated to whether a restart mode is used because DA is performed only once at the analysis time, regardless of the restart mode. The difference between a restart and a cold start mode only lies in the treatment of tendencies within the model and does not affect the analysis process. While a cold start mode initializes all the tendencies (as zeros) at the initial (or analysis) time, a restart mode carries them over from previous forecasts (e.g. nonzeros) for model integration. Also, as stated in L137-141, because all the prognostic variables are handled in flux form inside the model, they should be recoupled after data assimilation, but the recoupling also occurs only once at the analysis time, regardless of the restart mode (and even with IAU). No changes.

l229, what is YAML?

⇨ As defined in https://yaml.org/, YAML is a human-friendly data serialization language for all programming languages, which is basically an ascii file with its own writing style.

⇨ We've now added "(YAML is an ascii data format; https://yaml.org, last access: 27 Dec 2023)" in new L235. Thank you.

l225-235, is UFO an independent module outside of minimization or filtering? If so, how the bias correction (VarBC) is performed for radiance observations? How's the inter-channel correlations handled by the UFO?

⇨ These questions are irrelevant to our IAU work. For general information on VarBC in UFO, please refer to https://jointcenterforsatellitedataassimilation-jedi-docs.readthedocs-hosted.com/en/latest/inside/jedi-components/ufo/varbc.html. No changes.

l241, it is interesting to know whether the IAU functions well with inhomogeneous grids?

⇨ As for the IAU benefits on a variable-resolution mesh, Ha et al. (2017) reported that forecast errors were significantly reduced over the tropics (with both resolution-transition and high-resolution parts included). But it was not clear if the benefits were mainly tied to the mesh configuration or more affected by model errors or simulated flow regimes dominant over the region. No changes.

Section 4, it would be more convincing to have the statistical significance of the error differences between CTRL and IAU. It would be nice to have the verifications of CTRL and IAU relative to ECMWF or NCEP analysis, especially for water vapor.

⇨ Thank you for your suggestion. As stated in the last paragraph of Sections 1 and 4 (especially L71-74: "As a technical paper, ~"), however, this study focuses on our new implementation as a technical development paper and does not discuss the comprehensive characteristics of the system. As such, the results are only presented as a proof of concept, demonstrating the successful development. We leave a comprehensive analysis for future studies.

l280-285, please give some explanations for the error differences between the CTRL and IAU. Why IAU helps over the tropics but not over the polar regions. Is it possible this is due to the moving systems over the tropics (Ge et al. 2023 JAMES)?

⇨ Same as above. Fig. 7 is not meant for a thorough investigation or scientific discussions.

Since the IAU has a time filtering feature (L35), it might have played a positive role in simulating low-frequency modes dominant over the tropics. But given that the GFS analysis (used for the verification in Fig.7) also suffers from its own errors, we did not intend to discuss the errors *per se*. The figure was mainly shown to introduce post-processing capabilities in the system (L290-298). No changes.

---

## Author Comment (AC3)

==============================
Response to anonymous referee #2:
Citation: https://doi.org/10.5194/egusphere-2023-2299-RC2

Thank you very much for your thorough review, which helped us improve the draft for clarity. In response to your comments, L20 and L177 are updated, and one paragraph is added in L222-224. Typos are corrected as well. Please find our point-by-point response (in blue) to your valuable comments below.

This paper focuses on implementing Incremental Analysis Update (IAU) in MPAS-JEDI and evaluating its impact on model forecasts. IAU helps reduce initial imbalances in model forecasts caused by dynamical and physical balance issues during data assimilation, so model forecasts using IAU exhibit improved control over initial noise.

It's recommended to specify the control variables used in data assimilation and how to transform them into analysis increments.

⇨ The definition and the transformation of control variables are needed to avoid representing a static background error covariance (B) matrix explicitly. However, we used a pure ensemble-variational approach with no static error covariance, as already stated in L217-218. The static B is not even formulated in this draft since it was not available in MPAS-JEDI at the time of our IAU implementation. The introduction of control variables is important in the traditional variational approach, but it is not relevant to discuss them within the pure 3DEnVar context (with no static B). We decided not to include control variables here to stay focused on the IAU implementation, which only involves variable transformations between analysis and prognostic variables. It has nothing to do with the conversion from control to analysis variables, if any, which is done outside IAU. Thank you for your understanding.

I believe that an ensemble size of 20 is not sufficient for global analysis, despite using an ensemble with 100% background error covariance. It would be good to mention localization and ensemble spread inflation methods.

⇨ Thank you for your suggestion. Although this study is only meant to demonstrate the successful implementation of IAU, we agree with you that it would be good to mention the localization applied to the ensemble background error covariance. New lines are now added in 222-224 as "Due to the small ensemble size, we also apply the distance-based correlation function by Gaspari and Cohn (1999) using 1200 and 6 km as full-width radii for horizontal and vertical covariance localization, respectively."

⇨ However, the deterministic 3DEnVar updates a single analysis, for which we estimated the ensemble background error covariance based on ensemble forecasts from GEFS without ensemble inflation. As the inflation is not used, it is not mentioned here.

Regarding the ensemble Kalman filter used for the initial conditions of the ensemble forecast in the 3DEnVar system, clarification on its specifics would be beneficial for a comprehensive understanding.

⇨ This study did not employ ensemble Kalman filter, and ensemble forecasts were run from GEFS offline. As the 3DEnVar method used here only produces a single analysis (e.g., a single initial condition), this comment is not applicable to our study. But to emphasize the 3DEnVar algorithm used in this study, we added a new paragraph "For the minimization process, we employ an incremental approach (e.g., minimizing the cost function for increments) (Courtier et al. (1994)). in new L219-220.

In line 21 of the manuscript, there is an expression "It does not consider dynamical or physical balances across model grids or variables, nor does it account for the conservation of mass, momentum, or energy. Hence, the initial balance of the atmospheric flow can by disrupted by data assimilation when the initial state is replaced by the analysis state." While acknowledging that analysis increments from data assimilation might not fully reflect the model's balance, it's important to note that data assimilation does account for dynamical or physical balances across model grids or variables using background error covariance. For instance, temperature observations impact surrounding grids, affecting wind and humidity variables.

⇨ Thank you for your insight. You are correct that we can update unobserved states with observed variables in the DA procedure. But that can also be done through linear regressions, without imposing any dynamical or physical balance constraint, as often conducted in ensemble data assimilation. Here, we referred to the generic Kalman filter update, which is not specific to any particular DA algorithms or weather applications. You are also right in that the background error covariance is often estimated from sample forecasts (which employ governing equations for dynamics and physics), but that practice is primarily applied to variational approaches for atmospheric data assimilation. Please note that the particular way of estimating background error covariance is not required by the Kalman filter equation, or more generally, data assimilation. It is just one practical way of representing error statistics in numerical weather prediction models.
⇨ But in line with your perspective, we've now updated the statement as "It is not required to account for dynamical or physical balances across model grids or variables, nor does it ensure the conservation of mass, momentum, or energy.".

The authors notes at line 177 "it is easily extended for 4DIAU with varying weights over the IAU time window", However, transitioning from 3DIAU to 4DIAU may not be straightforward in the current version of MPAS-JEDI. Expanding to 4DIAU would necessitate multiple analysis increments over different times, requiring adjustments to the 4DEnVar and cycle suite.

⇨ We agree with you that 4DIAU would require some modifications associated with 4DEnVar and cycling scripts. In respect to your opinion, we've now changed "it is easily extended" to "it could be extended".

The following are minor correction requests.

⇨ Thank you for finding all the typos! Everything has been corrected except for the last comment. L240 (which is new L245) remains unchanged since 'cycling' itself is not considered plural.

L21: can by disrupted by data assimilation

-> can be disrupted by data assimilation

L30: the incremental analysis update (IAU) method was introduced in Bloom et al. (1996)

-> the incremental analysis update (IAU) method was introduced by Bloom et al. (1996).

L47: stability in the the fine mesh region.

-> stability in the fine mesh region.

L74: Details of the implementation is described

-> Details of the implementation are described

L110:  and the second term in the right-hand side

-> first and the second term on the right-hand side

L195: while ensuring the model forecasts reproducible

-> while ensuring the model forecasts are reproducible

or while ensuring reproducibility the model forecasts are

L240: global analysis and forecast cycling was conducted

-> global analysis and forecast cycling were conducted

---

## Author Comment (AC4)

Supplement Figures S1 – S4:

[Figure]

Figure S1. Vertical profile of water vapor mixing ratio (Qv [g/kg]) in 6-h forecasts from CTRL (black) and IAU (red), compared to the ERA-5 analysis, averaged over the cycles from April 21 to May 13, 2018. The mean error below 3 km is indicated next to each experiment name.

[Figure]

Figure S2. The Qv rms differences from the CTRL in percentages in IAU verified against radiosonde observations in a) the vertical profile at the analysis time and b) the time series of subsequent 24h forecasts of the corresponding metrics between 250 and 1000 hPa.

[Figure]

Figure S3. The global distribution of (a) the RMS errors of the CTRL run in 6-h background forecasts for surface pressure ([Pa]), verified against surface observations and (b) the RMS differences from the CTRL run in IAU.

[Figure]

Figure S4. The rms percentage differences from the CTRL in IAU for 6-h background forecasts in surface pressure across latitudes. The error bars denote the standard deviation, corresponding to a 95% confidence level.

---

## Author Response (AR2)

Response to the Editor Shu-Chih Yang:
Citation: https://doi.org/10.5194/egusphere-2023-2299-EC1

Thank you very much for your own review. In response to your comments, we provide four figures in the supplement as Figs. S1-S4, adding a few paragraphs in the manuscript (Page 10). Please find our point-by-point response to your valuable comments below. Our responses are marked in blue.

Although the authors claim that the manuscript is categorized as a "technical" paper, the associated implementation is successful with **comprehensive** justification. I agree with the reviewer's comments about showing the results related to water vapor since it is vital to know whether implementing the IAU method affects the moisture accuracy of MPAS-A.

⇨ We totally agree with you that any new implementation should be thoroughly investigated for correctness. For that, we have cross-checked all mathematical expressions and their corresponding codes multiple times. However, we want to take this opportunity to clarify that ensuring successful development is distinct from providing comprehensive justification for the impact study, which involves various scientific aspects. We also note that the main motivation and purpose of the IAU implementation are to effectively suppress initial noise resulting from dynamic imbalances, as already demonstrated in Figs. 2 and 3.

⇨ But we agree that it would be interesting to examine the impact of IAU on moisture as part of the control variables. In response to your comment, we provide two figures below.

⇨ First, we compare water vapor mixing ratio (Qv [g/kg]) in 6-h forecasts between CTRL and IAU against ERA-5 analysis globally over a total of 92 6-hourly cycles from April 21 to May 13, 2018 (e.g., after a one-week spin-up period). Although root-mean-square errors are the same as 0.7 in both experiments (not shown), the systematic bias indicates that IAU produces slightly better agreement with ERA-5 analysis than CTRL, especially in the boundary layer (< 2 km).

[Figure]

Figure S1. Vertical profile of water vapor mixing ratio (Qv [g/kg]) in 6-h forecasts from CTRL (black) and IAU (red), compared to the ERA-5 analysis, averaged over the cycles from April 21 to May 13, 2018. The mean error below 3 km is indicated next to each experiment name.

⇨ We also present a panel plot below, where we verify the analysis and the first-day forecast against sounding observations between the surface and 250 hPa for the entire month. Compared to the CTRL run, RMS errors are slightly worse at the analysis time (which corresponds to a 3-h forecast in the IAU run, as opposed to the analysis in CTRL) by up to 2% in (a). But in (b), as forecasts start from the analysis, the errors decrease during the first 6 h, and exhibit a statistically significant reduction of approximately 4% at 18 h in the troposphere. Hence, we cannot claim that IAU consistently improves moisture analysis and forecasts, but it is fair to say that it does lead to some improvements.

[Figure]

Figure S2. The Qv rms differences from the CTRL in percentages in IAU verified against radiosonde observations in a) the vertical profile at the analysis time and b) the time series of subsequent 24h forecasts of the corresponding metrics between 250 and 1000 hPa.

⇨ As we believe that it would be best to stay focused on surface pressure to represent the entire column, we have decided to include them in the Supplemental material, with the comment in the text (L282) as "The moisture verification for the 6-h forecast is provided in the supplement.". Thank you for your understanding.

In addition, it is worth addressing the issue of the degradation of the northern polar region with IAU. According to Fig. 7b, the rms difference is about 20% in the north polar region and 10% near 25-degree S during the first-day forecast! If the authors attributed the difference to their own error in GFS analysis, I suggest showing the same figure with other analyses as the reference, like the EC analysis.

⇨ Thank you for your careful review on Figure 7. We admit that it is challenging to explain the forecast degradation in a specific region, especially given that we did not find anything particularly suspicious in Fig. 3. We fully understand the editor's concern, though, and we would like to examine the issue through the observation-space diagnostics, rather than the verification against any particular analysis. This work was meant for the analysis cycling (with our own data assimilation), so it is legitimate to check the performance with respect to observations. In Figure S3, where we verified our background forecasts for surface pressure against measurements, we noticed one red dot near the North Pole (to the north of Greenland) in the bottom panel. And there are more red dots to the north of Russia, all of which seemed to contribute to the larger errors shown in Fig. 7b. Also, you correctly

captured the red area near 25°S, which seems associated with the red dots in the southern part of the tropics, as depicted in Fig. S3b. But as you can see through the global distribution, we do not have enough observations near the Poles and over the ocean to sufficiently constrain the model state. The month-long error statistics reveal almost no colors (e.g., little deviations) over the CONUS domain with a dense observing network, indicating that IAU itself does not degrade forecast errors when the model states are well constrained through data assimilation. It remains unclear whether the impact of IAU has intertwined with model errors in certain data-sparse regions. Note that model errors are not accounted for in this hybrid 3D-EnVar framework.

[Figure]

Figure S3. The global distribution of (a) the RMS errors of the CTRL run in 6-h background forecasts for surface pressure ([Pa]), verified against surface observations and (b) the RMS differences from the CTRL run in IAU.

⇨ It is worth noting that *we implemented IAU on the model's unstructured mesh in a generic way*, not specifically tied to geographic regions. For clarity, following the statement "Compared to the GFS analysis, MPAS forecasts in CTRL exhibit the largest (or the fastest) error growth in the Southern Hemisphere. Forecasts in the IAU run, on the other hand, tend to reduce errors in the tropics while increasing errors near the North Pole region.", we added a paragraph in L289-295, stating "This aligns with the findings of Ha et al. (2017), where forecast errors were significantly reduced over the tropics in a variable-resolution mesh, including both resolution-transition and high-resolution parts. Because IAU is implemented on the model's unstructured mesh (which is in a random order), it is not associated with particular geographic locations or mesh configurations. Given its time filtering feature, IAU might be more effective in simulating low-frequency modes dominant over the tropics. It is also noted that the impact of IAU may be nonlinearly intertwined with model errors in data-sparse regions, such as the Poles. However, model errors are not accounted for in the hybrid 3DEnVar system used in this study. Additional area-specific features in the verification are provided in the supplemental material.".

⇨ Fig. S4 further supports our point that the IAU significantly improved forecast errors in most regions, except for the North Pole area and the Southern Hemisphere Ocean, in a statistically significant manner. As illustrated in Fig. 4 in the manuscript, the IAU run assimilated slightly more observations (by 1-2%) throughout the month-long cycling, which is a good indicator that the DA cycling system work more effectively. Overall, it is our belief that our IAU implementation was successfully completed, with positive impacts on the analysis and short-term forecasts.

[Figure]

Figure S4. The rms percentage differences from the CTRL in IAU for 6-h background forecasts in surface pressure across latitudes. The error bars denote the standard deviation, corresponding to a 95% confidence level.